# Linear Capacitive Pressure Sensor with Gradient Architecture through Laser Ablation on MWCNT/Ecoflex Film

**DOI:** 10.3390/polym16070962

**Published:** 2024-04-02

**Authors:** Chenkai Jiang, Bin Sheng

**Affiliations:** 1School of Optical-Electrical and Computer Engineering, University of Shanghai for Science and Technology, Shanghai 200093, China; 15003438539@163.com; 2Shanghai Key Laboratory of Modern Optical Systems, Engineering Research Center of Optical Instruments and Systems, Shanghai 200093, China

**Keywords:** flexible capacitive pressure sensor, gradient micro-cone architecture, laser ablation, dielectric constant, linear response, silicone rubber

## Abstract

The practical application of flexible pressure sensors, including electronic skins, wearable devices, human–machine interaction, etc., has attracted widespread attention. However, the linear response range of pressure sensors remains an issue. Ecoflex, as a silicone rubber, is a common material for flexible pressure sensors. Herein, we have innovatively designed and fabricated a pressure sensor with a gradient micro-cone architecture generated by CO_2_ laser ablation of MWCNT/Ecoflex dielectric layer film. In cooperation with the gradient micro-cone architecture and a dielectric layer of MWCNT/Ecoflex with a variable high dielectric constant under pressure, the pressure sensor exhibits linearity (R^2^ = 0.990) within the pressure range of 0–60 kPa, boasting a sensitivity of 0.75 kPa^−1^. Secondly, the sensor exhibits a rapid response time of 95 ms, a recovery time of 129 ms, hysteresis of 6.6%, and stability over 500 cycles. Moreover, the sensor effectively exhibited comprehensive detection of physiological signals, airflow detection, and Morse code communication, thereby demonstrating the potential for various applications.

## 1. Introduction

The wide use of flexible sensors, including use in applications such as electronic skins [1,2,3,4], human–computer interaction [5,6], cardiovascular monitoring [7], body joint detection [8,9,10], breathing tests [11,12,13], and information communication [14,15], has attracted academic attention. According to the working principles, flexible pressure sensors can be classified as piezoresistive [16,17], capacitive [18], triboelectric [19], and piezoelectric [20], which can effectively convert mechanical deformation into quantifiable electrical signals. However, the main problem with existing pressure sensors is the nonlinear change in sensor sensitivity under pressure, making it imperative to devise a capacitive pressure sensor with both good sensitivity and linearity.

Solid silicone rubber (PDMS, Ecoflex) is often used in conventional capacitive sensors, but the sensitivity is limited due to the lack of a micro-structured dielectric layer and poor deformation capacity. Therefore, microstructure dielectrics with higher compressibility have been developed to raise sensitivity, including micro-domes [21], micro-cones [22], micro-pyramids [23,24,25], and porous structures [26,27,28]. Although microstructure dielectrics can be manufactured using various molds [29,30], the expense and inefficiency of the technology limits its development. To overcome these constraints, Xue et al. [31] introduced the method of obtaining microstructures in dielectric layers through laser ablation, which is of high efficiency and which is capable of producing large manufacturing areas, providing a new way to prepare microstructures.

In addition, several studies have already proved that increasing the dielectric constant can improve the sensitivity of sensors [32,33,34], which is usually achieved by mixing silicone rubber with conductive fillers (carbon nanotubes [33] and graphene [35]). Meanwhile, carbon nanotubes, due to their excellent conductivity and high aspect ratio, can significantly reduce dielectric permeation thresholds and have become the most commonly used conductive filler in research.

To improve the linearity of the sensor over the working range, Zhou et al. [36] successfully presented a capacitive sensor with the MWCNT/PDMS gradient architecture dielectric layer, achieving a high level of linearity (R^2^ = 0.993) and sensitivity of 0.065 kPa^−1^ at a pressure of 1600 kPa. Xie et al. [37] proposed a linear capacitive pressure sensor with wrinkled PDMS/CNT dielectric layers that exhibits good linearity (R^2^ = 0.975) and sensitivity of 1.448 kPa^−1^ at 20 kPa. However, the above studies failed to analyze the variations in the dielectric constant of dielectric layer composite films when film is subject to pressure. This crucial factor ensures linearity response in pressure sensors, which is the issue explored in our study.

In this work, we present an MWCNT/Ecoflex pressure sensor with a gradient micro-cone architecture achieved through laser ablation. The response in the relative dielectric constant of the MWCNT/Ecoflex film under pressure was investigated. The sensor exhibited good linearity and sensitivity by coordinating the gradient micro-cone architecture and the film, reaching sensitivity of 0.75 kPa^−1^ and significantly improving linearity (R^2^ = 0.990) with the 2.5 wt% MWCNT under 0–60 kPa. Additionally, the sensor demonstrated excellent repeatability with a fast response time of 95 ms and a recovery time of 129 ms over 500 cycles. Laser ablation significantly accelerates the microstructure manufacturing process, and the gradient micro-cone-structured dielectric layer can be formed within 5 min, which supports being customized separately. Finally, we demonstrate the applications of pressure sensors in human signal detection, airflow detection, and Morse code communication.

## 2. Materials and Methods

### 2.1. Materials

MWCNTs (purity: >95%; diameter: <8 nm; length: 0.5–2 µm) were purchased from Suzhou Tanfeng Material Technology. Ecoflex 00-30 was obtained from Smooth-On. Hexane (≥97.0%) purchased from the China Pharmaceutical Group Corporation was used as the diluent.

### 2.2. Preparation of Ecoflex Film

The beaker, glass rod, and Petri dish were cleaned with hexane and then dried in a heated oven at 60 °C for 30 min. Subsequently, they were removed from the oven and cooled. Ecoflex parts A and B were mixed at a ratio of 1:1 and stirred for 15 min. The resulting mixture was then poured into a Petri dish and put into a vacuum oven for 15 min to eliminate air bubbles. The Ecoflex fluid was then cured at room temperature for 3 h.

### 2.3. MWCNT/Ecoflex Film Preparation

The MWCNT and dispersant (polyvinylpyrrolidone, PVP) were precisely quantified to provide a mass ratio of 10:1 [38]. The MWCNT and PVP were dispersed into the hexane solution by an ultrasonic cleaner (30 min) (GW0303, GW Ultrasonic Instruments Co., Ltd., Shenzhen, China). Ecoflex part A was then incorporated into the solution and agitated at 1000 rpm for 30 min at high speed. Ecoflex part B was then added and the mixture was stirred for an additional 30 min. The resulting liquid mixture was poured into a mold. After the evaporation of hexane, the uncured MWCNT and Ecoflex mixture was then vacuum-dried in an oven for 30 min and heated at 60 °C for 1 h to achieve complete curing.

### 2.4. Experimental Setup

The sensor was placed on a pressure test platform (ZQ-990B, ZhiTuo, Ltd., Dongguan, Guangdong, China) that is capable of applying a pressure of 0–60 kPa. The capacitance data were collected using an LCR instrument (TH2822D, Tonghui, Jiangsu, China) with a copper wire connecting the sensor and the LCR operating at 1 kHz. All data were collected at a room temperature of 25 °C. The morphologies of the samples were analyzed using a scanning electron microscope (JSM-IT500HR, Japan Electronics, Tokyo, Japan). Fourier Transform Infrared (FTIR) spectra of the samples were obtained using a Spectrum FTIR spectrometer manufactured by PerkinElmer (L1280127, PerkinElmer, Waltham, MA, USA). X-ray photoelectron spectroscopy (XPS) (AXIS Ultra DLD, Shimadzu, Kyoto, Japan) was used to analyze the elemental composition of samples. Raman spectra were acquired through use of a Raman microscope (RAMANtouch, Nanophoton, Kyoto, Japan) with 532 nm laser excitation. The Tri-Strong TIDE Industrial Camera Microscope was used to measured optical microscope images.

## 3. Results and Discussion

### 3.1. Fabrication of the Sensor

Figure 1a shows the structure of the sensor. The sensor comprises electrodes and a gradient micro-cone architecture dielectric layer. It can be attached to the skin to measure physiological parameters. Among them, the electrodes and dielectric layer were glued together with Ecoflex. Copper foil (5 μm) and PI tape (0.02 mm) were combined to form the electrodes. The fabrication schematic for the dielectric layer is shown in Figure 1b. First, the MWCNT/Ecoflex solution was poured into a mold for curing. Second, an 8 × 8 (8 mm × 8 mm) array was designed using CAD and downloaded to a CO_2_ laser (K3020, Julong Laser Co, Ltd., Liaocheng, Shandong, China). Finally, we used a CO_2_ laser to ablate the surface of the MWCNT/Ecoflex film to form a gradient micro-cone architecture after cleaning the carbide material produced during the ablation process. The finished sensor is shown in Appendix A.

To investigate the effect of microstructures on sensor performance, the Ecoflex film was ablated using different power levels and different numbers of ablations. The ablation region is shown in black in Appendix A, in which an 8 × 8 array of micro-cones was fabricated. The low heights of the micro-cones (Appendix A) can be attributed to the utilization of low power when we ablated the films using the parameters in Appendix A. Conversely, when the Ecoflex film was ablated utilizing the parameters in Appendix A, the high power destroyed the micro-cones (see Appendix A). Consequently, we used the appropriate power parameters in Appendix A for the following study, which allowed us to ablate high micro-cones (see Appendix A) with increasing micro-cone heights as the number of ablations increased.

Figure 2a shows the relative capacitive response of the Ecoflex dielectric sensor with different numbers of ablations at 0–60 kPa. Sensor sensitivity increases as the number of ablations increases, owing to the improved deformability of the dielectric layer caused by the increased height of the micro-cones. Sensor sensitivity was 0.041 kPa^−1^ at 0–60 kPa when the ablations were repeated four times. As described in [36], it has been shown in recent years that sensors with linear gradient architecture dielectric layers exhibit more linearity and sensitivity than those without a gradient architecture dielectric layer. Therefore, to fabricate a micro-cone dielectric layer with linear gradient micro-cones, small areas were ablated (Appendix A) after the micro-cones were fabricated, as shown in Appendix A. Finally, four levels of linear gradient micro-cones (see Appendix A) were fabricated in the dielectric layer. Two ablation schemes with different numbers of height distributed micro-cones are shown in Appendix A. Figure 2b shows the sensor’s relative capacitive response for the three ablation schemes. The sensitivity of the ablation scheme shown in Appendix A (0.055 kPa^−1^) is higher than that of the scheme shown in Appendix A (0.047 kPa^−1^) and the scheme shown in Appendix A without the linear gradient architecture (0.041 kPa^−1^). The linear range of Appendix A is also greater than that of Appendix A without a gradient architecture. In summary, we chose Appendix A for the ablation in the present study. However, sensor sensitivity still dropped, resulting in a nonlinear response over the pressure range (0–60 kPa). To solve these issues, Ecoflex film was mixed with MWCNTs.

### 3.2. Characterization

The height of the micro-cone ablation for MWCNT/Ecoflex was slightly lower than that of the micro-cone ablation for Ecoflex alone. Figure 3a–d show side views of the MWCNT/Ecoflex micro-cones. The height of the micro-cone structure increases as the number of ablations increases (see Appendix A). Figure 3e,f show SEM images of the MWCNT/Ecoflex dielectric layer with gradient micro-cones, enabling distinct observations of the laser ablation micro-structured at four different height levels (see Appendix A). Figure 3f–i show the SEM images of the cross-section of the MWCNT/Ecoflex film. The presence of micron-scale porous cracks in the cross-section, which has a porosity of 2.7%, was attributed to the volatilization of hexane.

Figure 4 displays the ATR-FTIR and Raman spectra, and Figure 4a illustrates the ATR-FTIR spectrum acquired for the MWCNT/Ecoflex film, the MWCNT/Ecoflex film after laser ablation, and the ablation product of the MWCNT/Ecoflex film. The absorption peaks of the MWCNT/Ecoflex film are almost consistent with those of the MWCNT/Ecoflex film after laser ablation because the Si-C and SiO_2_ peaks are located at a position similar to that of Si-O-Si, and the signals of its peak are hidden in the Si-O-Si peak. The spectrum exhibits an absorption peak at 2963 cm^−1^, which can be attributed to the asymmetric vibration of CH_3_. The CH_3_ group exhibited an asymmetric stretching vibration, leading to a relatively weak peak at 1412 cm^−1^ and a strong absorption peak at 1258 cm^−1^. The asymmetric stretching vibration and deformation of Si-O-Si caused significant absorption peaks at 1066 cm^−1^ and 1010 cm^−1^. The prominent absorption peak at 788 cm^−1^ was related to Si-CH_3_ [15]. The ATR-FTIR spectra of the ablation product of MWCNT/Ecoflex film are depicted in Figure 4a, which shows an absorption peak at 1066 cm^−1^ from the asymmetric tension vibration of Si-O-Si. The weak peak at 800 cm^−1^ was related to the asymmetric and symmetric deformation of Si-O and Si-C. Therefore, we considered that SiO_2_ and SiC were produced during laser ablation. These analyses above were similar to the findings of previous studies [15,39]. In addition, Figure 4b shows the Raman spectra of the Ecoflex film, MWCNT film, and the MWCNT/Ecoflex film, as well as the MWCNT/Ecoflex film after laser ablation. Here, the MWCNT film, MWCNT/Ecoflex film, and laser-ablated MWCNT/Ecoflex film showed three peaks of carbon in the D band at ~1346 cm^−1^, the G band at ~1582 cm^−1^, and the 2D band at ~2703 cm^−1^. Meanwhile, for the MWCNT/Ecoflex film, the laser-ablated MWCNT/Ecoflex film also exhibited the characteristic peaks of the Ecoflex signals, and the Si-C signals of the laser-ablated films are adjacent to the peaks of the Ecoflex signals. These peaks are consistent with the conclusions of a previous study [40].

To comprehensively analyze the composition and functional groups present in MWCNT/Ecoflex film, the XPS spectra of C1s, O1s, and Si 2p in these films were collected. The fitted spectra of the C1s peaks of the MWCNT/Ecoflex film and the laser-ablated MWCNT/Ecoflex film (Figure 5a,d) can be deconvolved into two distinct peaks at 284.4 and 284.8 eV, which are attributed to C-C(sp2) and C-C(sp3). The fitted spectra of the O1s peaks of the MWCNT/Ecoflex film and the laser-ablated MWCNT/Ecoflex film (Figure 5b,e) exhibit two distinct peaks at 531.9 eV and 532.7 eV, signifying C-Si-O and Si-O_x_. The fitted spectra of the Si 2p peaks of the MWCNT/Ecoflex and laser-ablated MWCNT/Ecoflex films (Figure 5c,f) can be decomposed into three distinct peaks at 101.8 eV, 102.5 eV, and 100.9 eV, respectively, which are attributed to C-Si-O, Si-O_x_, and Si-C. Notably, the x values of the Si-O_x_ peaks in the O 1s and Si 2p spectra of the MWCNT/Ecoflex film and the MWCNT/Ecoflex film after laser ablation are estimated to be in the range of 1 to 2. These peaks are consistent with previous findings [40,41,42]. The XPS findings are consistent with the findings obtained from the Raman and ATR-FTIR spectroscopy analyses.

The dielectric constant of the MWCNT/Ecoflex film was also investigated as shown in Equation (1), where $\varepsilon_{0},{\;\varepsilon }_{r},\;A,\;\mathrm{and}\;d$ represent the dielectric constant in vacuum, the relative dielectric constant, the area directly opposite electrodes, and the distance between the electrodes, respectively. c is the capacitance. Figure 6a shows the dielectric constant of the MWCNT/Ecoflex film at different mass fractions. The MWCNT/Ecoflex film exhibited an optimal relative dielectric constant of 650 at 2.5 wt%.
(1)$$\varepsilon_{r}=\frac{cd}{\varepsilon_{0}A}$$
(2)$$\sigma =\frac{l}{RA}$$

The conductivity and percolation thresholds are shown in Figure 6b. The equation for electrical conductivity is outlined in Equation (2). σ represents conductivity, R denotes the resistance of the MWCNT/Ecoflex film, A represents the film’s cross-sectional area, and l indicates the spacing between electrodes. Owing to the exceptional electrical conductivity and high aspect ratios exhibited by MWCNTs, the film was able to swiftly establish conductive pathways. Hence, percolation theory exhibits a significant increase in conductivity at the critical mass fraction of MWCNTs. Percolation theory establishes the relation between conductivity and the mass fraction of MWCNTs as follows:(3)$$\sigma_{a}\propto \sigma_{0}{\left(p-p_{a}\right)}^{t}$$

As shown in Equation (3), $\sigma_{a}$ represents the film’s conductivity and $\sigma_{0}$ is the initial conductivity, where p denotes the MWCNT mass fraction, $p_{a}$ signifies the percolation threshold, and t is the threshold index. For the MWCNT/Ecoflex films, Equation (3) was used to calculate $p_{a}$ and t as 0.7 wt% and 2.4, respectively. These data are similar to the findings of previous studies [43,44]. Figure 6c illustrates the Young’s modulus of the MWCNT/Ecoflex films with different mass fractions of MWCNTs, ranging from 0.40 MPa to 0.54 MPa. Figure 6d depicts the response in permittivity under pressure (0–60 kPa) with different mass fractions of the MWCNT/Ecoflex films. We collected the value of the deformation of the film under pressure on the test platform, and the initial thickness value minus the deformation value is the true height of the film under pressure. Using Equation (1), we then obtained the exact dielectric constant of the film under pressure. All of the above capacitance data were measured at an LCR frequency of 1 kHz. The response of the relative dielectric constant of the films increases with increasing MWCNT mass ratio under 60 kPa and reaches a maximum 3.2 at 2.5 wt%.

### 3.3. The Performance of the Flexible Pressure Sensor

A classical parallel plate capacitor was used for our capacitive pressure sensor, as shown in Equation (4). In our sensor, the dielectric constant under vacuum and the area directly opposite the area directly opposite the two electrodes was fixed. In this case, the capacitance change is caused by $\varepsilon_{r}\;$ and d. Figure 7a shows a working flow diagram and the equivalent circuit.
(4)$$c=\frac{\varepsilon_{0}\varepsilon_{r}A}{d}$$

The sensor’s dielectric layer consists of the micro-cone layer and the thin film layer. The micro-cone layer contains two parallel capacitors denoted by $c_{1\;}$ and $\;c_{2}$, where $c_{1}$ represents a capacitor with air in the micro-cone layer and $c_{2}$ represents the MWCNT/Ecoflex micro-cones. The MWCNT/Ecoflex dielectric film capacitor is indicated by ${\;c}_{3}$. As shown in Figure 7a-I, the three capacitors maintain their initial capacitance under unpressured conditions. In Figure 7a-II, it can be observed that the gradient architecture micro-cones in the sensor are gradually compressed when subjected to pressure. This can be attributed to the air and lower Young’s modulus in the micro-cone layer compared to the film layer, while the Δd value of the micro-cone layer exhibits a rapid increase during this stage. Meanwhile, the Δd value of the film layer is almost negligible. Equation (5) shows the Lichtenecker rule, where $\varepsilon_{a}$, ${\varepsilon \;}_{c}$, ${\;v}_{a},$ and $v_{c}$ represent the relative dielectric constant of air, the relative dielectric constant of the MWCNT/Ecoflex film, and the volume fractions of air and the MWCNT/Ecoflex film, respectively. When the sensor’s dielectric layer is compressed, the air is replaced by the MWCNT/Ecoflex material. In this study, the dielectric constant of $\varepsilon_{a}$ was equal to 1, while the dielectric constant of $\varepsilon_{c}$ for the 2.5 wt% MWCNT/Ecoflex composite was measured to be 650. As the sensor is compressed, the MWCNT/Ecoflex micro-cones progressively replace the air, resulting in an increase in the relative dielectric constant, which is indicated by $\varepsilon_{t}$. In Figure 7a-II, the sensor’s response to pressure is related to changes in $c_{1}$ and ${\;c}_{2}$ in the micro-cone dielectric layer, while the change in $c_{3}$ is negligible. The micro-cone layer was pressed into the film as the pressure increased, as shown in Figure 7a-Ⅲ. The response for $c_{3}$ is that of sensor capacitance. As the dielectric constant of the film increases under pressure, the capacitance of the sensor continues to increase. Therefore, the micro-cone dielectric layer, in combination with a thin film, results in a sensor with high sensitivity and linearity.
(5)$$\varepsilon_{t}=\varepsilon_{a}v_{a}+\varepsilon_{c}v_{c}$$
(6)$$s=\frac{𝜕\left(\frac{\Delta c}{c_{0}}\right)}{𝜕p}$$

Optimization of dielectric layer film thickness is also considered (Appendix A). When the thickness of the film layer is 200 μm, the compression range of the dielectric layer is smaller, resulting in a working range that is narrower than that of a sensor with a thickness of 400 μm. When the thickness of the film layer reaches 600 μm, the linear range of the sensor is smaller than that of a sensor with a substrate thickness of 400 μm, which means that the microstructures become less useful in the pressure range when the thickness of film increases. This result is consistent with the findings of previous related work [23]. Figure 7b illustrates the relative capacitive response of the pressure sensor with different mass fractions of MWCNTs at 60 kPa. The relative capacitance change is expressed as $\Delta c/c_{0\;}$, in which Δc and $c_{0\;}$ represent the change capacitance and the starting capacitance, respectively, while 𝜕p is the range of pressure changes. Compared to the other MWCNT mass fractions, the relative dielectric constant reaches an optimum value of 650 at 2.5 wt% (see Figure 6a). Meanwhile, the variation in the MWCNT/Ecoflex film’s relative dielectric constant under pressure also shows the highest change at 2.5 wt% (see Figure 6d). These two factors are responsible for the highest response of the sensor occurring at 2.5 wt%. The sensitivity of the sensor is defined by Equation (6). When the mass fraction of MWCNTs was 2.5 wt%, the capacitive sensor exhibited a sensitivity 0.75 kPa^−1^ at 60 kPa, which is 14 times higher than that with the Ecoflex dielectric layer. The linearity of the sensor (R^2^ = 0.990) is shown in Figure 7c. As shown in Figure 7d, a maximum hysteresis of 6.6% is obtained by loading and unloading pressure. Figure 7e compares the performance of our sensor and other capacitive sensors in terms of sensitivity and linearity. Detailed information is provided in Appendix A. Figure 7f displays the capacitance response of the sensor when a step pressure of 0–60 kPa was applied to it. The limit response of the sensor at 1 Pa was also investigated (Appendix A). Figure 7g shows a response time of 95 ms and a recovery response time of 129 ms at a pressure of 1.3 kPa. Due to its close proximity to human skin’s response time, it can be applied to applications involving real-time signal processing on the surface of humans [45]. To examine the stability of the sensor under pressure, a 500-cycle test was performed at 60 kPa, as shown in Figure 7h where 10 cycles between 90 and 100 cycles and 420 and 430 cycles are shown.

### 3.4. Applications of the Sensor

Pulse signals indicate an individual’s health because of their association with cardiac systole and diastole. Our sensor can detect small signals from carotid arteries, as shown in Figure 8a. Swallowing is a complicated muscular process in the human body, and it is noteworthy that various digestive disorders can impede swallowing. Consequently, the monitoring of swallowing signals becomes imperative in the course of treatment for such disorders. In Figure 8b, throat swallowing is monitored by attaching the sensor. Figure 8c shows that the sensor can detect a bending elbow. The capacitive signal, which increases with the degree of elbow flexion, also increases. This indicates that the sensor has the potential to be a reliable instrument for recording arm activities. Figure 8d shows the pressure sensor attached to the body’s abdomen. It can be observed that the capacitance changes regularly with breathing. The sensor can thus detect muscle activity. Assessment of the condition of the calf muscles is imperative for sprinters during the development of a scientific training program. The sensor can be affixed to the calf region, where it monitors the tension and relaxation of the calf muscles, as shown in Figure 8e. Additionally, the sensor was able to detect the perceived degree of airflow (Figure 8f). Morse code is a form of communication that can be emitted not only by radio but also by other means, such as sound and gestures. This type of communication plays an irreplaceable role in emergency rescue and disaster preparedness. Figure 8g–i show an application of Morse code, where the user presses a finger on the sensor causing a change in capacitance, which outputs the Morse code for “SOS” and “USST”.

## 4. Conclusions

In this work, we proposed the fabrication of a new manufacturing method for capacitive pressure sensors whose MWCNT/Ecoflex dielectric layer includes a thin film and a gradient micro-cone structure achieved by CO_2_ laser ablation. This fabrication method offers the advantages of high manufacturing efficiency and customizability. We have optimized the power and speed of the laser ablation to determine the optimum ablation parameters, as well as the optimal height of the microstructure. Variation in the MWCNT/Ecoflex film’s relative dielectric constant under pressure and with different mass fractions was also investigated. In addition, we studied the effect of doped MWCNTs on sensor performance and ultimately confirmed that the optimal configuration was MWCNTs with a mass ratio of 2.5 wt%. The sensor exhibited a sensitivity of 0.75 kPa^−1^ and demonstrated exceptional linearity (R^2^ = 0.990) from 0 to 60 kPa with a response time of 95 ms, a low lag time of 6.6%, and a recovery time of 129 ms. Finally, this study elucidates the potential applications of this sensor, including the detection of body signals such as pulse signals, swallowing signals, and arm joint flexion signals, the testing of calf muscle strength, and outputting Morse code messages entered by finger presses.

## Figures and Tables

**Figure 1 polymers-16-00962-f001:**
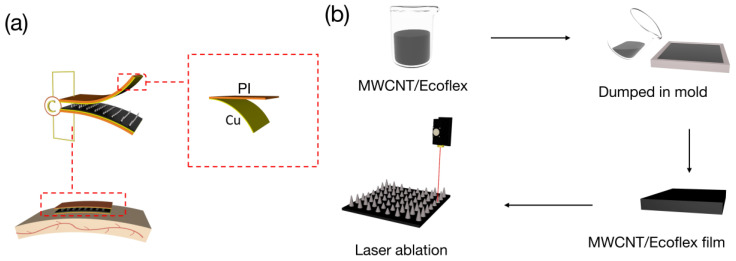
(**a**) Structure of a flexible pressure sensor on human skin. (**b**) Dielectric layer fabrication step for flexible pressure sensors.

**Figure 2 polymers-16-00962-f002:**
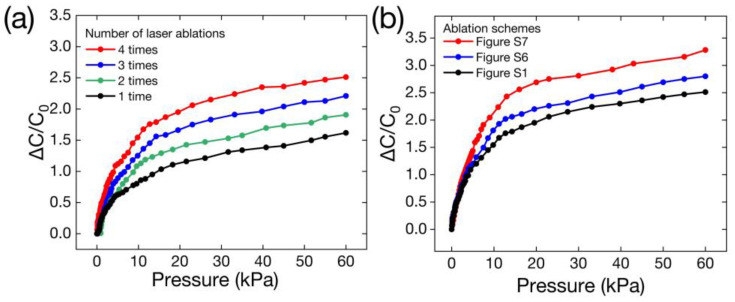
Relative capacitive response of the Ecoflex dielectric pressure sensor under 0–60 kPa. (**a**) Relative capacitive response of the Ecoflex pressure sensor with different numbers of ablations. (**b**) Relative capacitive response of Ecoflex pressure sensors with three different laser ablation schemes.

**Figure 3 polymers-16-00962-f003:**
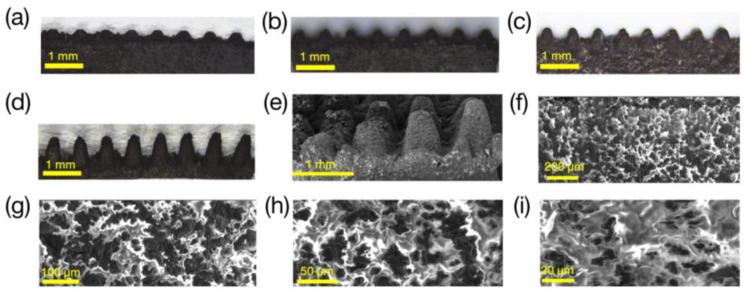
The surface morphology of MWCNT/Ecoflex dielectric layers. (**a**–**d**) Side view optical images of the MWCNT/Ecoflex micro-cones obtained using the parameters of Appendix A with different ablation numbers: (**a**) one time, (**b**) two times, (**c**) three times, (**d**) four times. (**e**,**f**) SEM images of the side view of the dielectric layer with gradient micro-cones. (**g**–**i**) Cross-sectional SEM images of the MWCNT/Ecoflex film. The black area is MWCNTs and the white area is Ecoflex.

**Figure 4 polymers-16-00962-f004:**
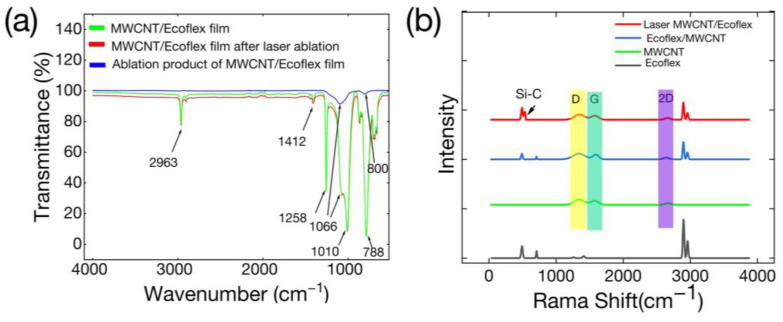
The ATR-FTIR spectrum and Raman spectrum. (**a**) The ATR-FTIR spectrum of MWCNT/Ecoflex film, the ATR-FTIR spectrum of MWCNT/Ecoflex film after laser ablation, and the ATR-FTIR spectrum of the ablation product of MWCNT/Ecoflex film. (**b**) The Raman spectrum of the MWCNT/Ecoflex film after laser ablation, the Raman spectrum of MWCNT/Ecoflex film, the Raman spectrum of MWCNT film, and the Raman spectrum of Ecoflex film.

**Figure 5 polymers-16-00962-f005:**
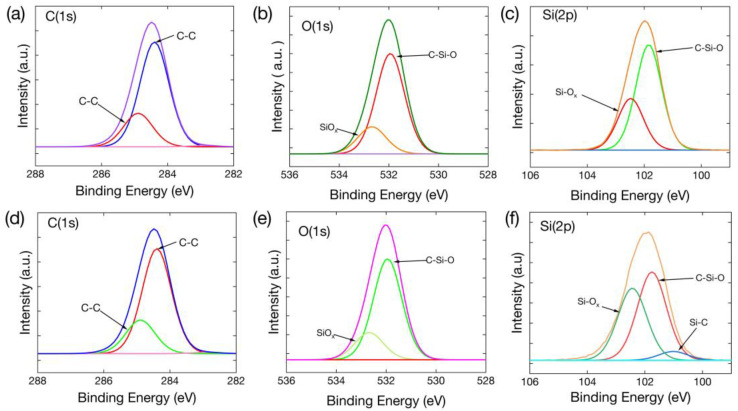
XPS spectra of the films. (**a**–**c**) XPS spectra of MWCNT/Ecoflex film for C 1s, O 1s, and Si 2p. (**d**–**f**) XPS spectra of MWCNT/Ecoflex film after laser ablation for C 1s, O 1s, and Si 2p.

**Figure 6 polymers-16-00962-f006:**
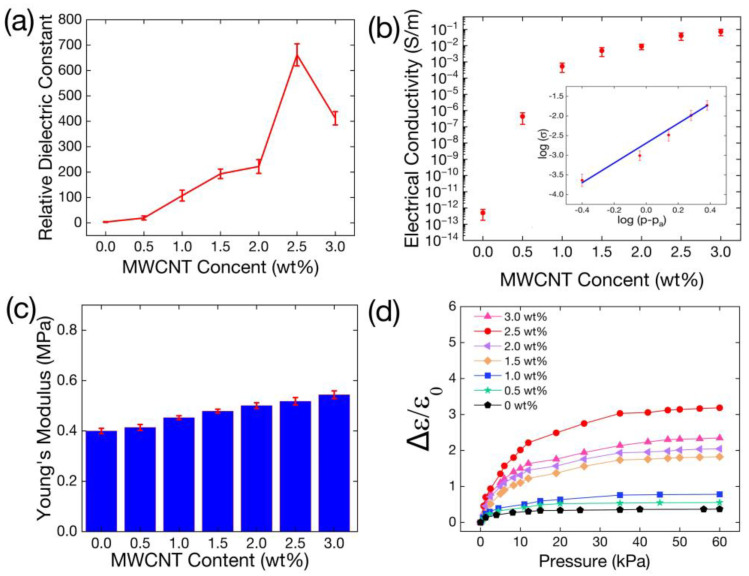
The characteristics of MWCNT/Ecoflex film. (**a**) Relative dielectric constant of MWCNT/Ecoflex films with different mass fractions of MWCNTs. (**b**) The relationship between the mass fraction of MWCNTs and the conductivity of MWCNT/Ecoflex films. (**c**) Young’s modulus of MWCNT/Ecoflex films with different mass fractions of MWCNTs. (**d**) The response of the relative dielectric constant of the MWCNT/Ecoflex films with different mass fractions of MWCNTs under 60 kPa.

**Figure 7 polymers-16-00962-f007:**
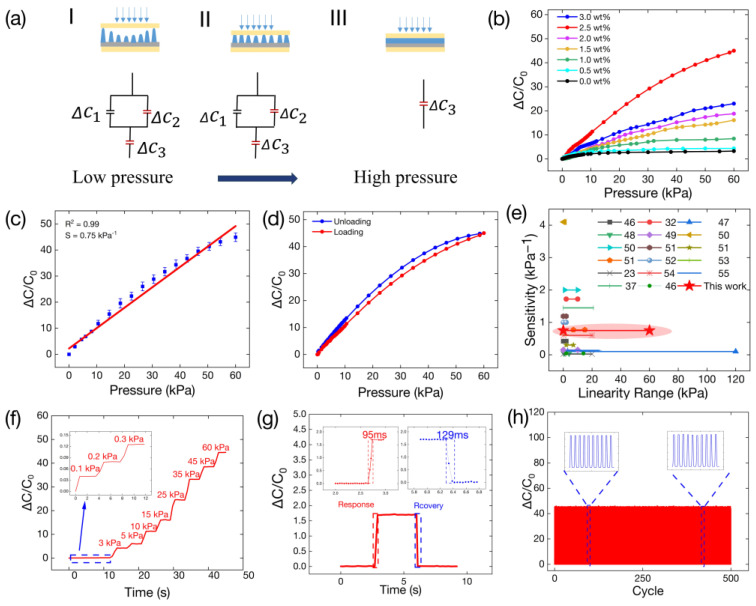
A working flow diagram and the performance of the sensor. (**a**) Working flow diagram of the MWCNT/Ecoflex capacitive pressure sensor with gradient architecture. (**b**) Relative capacitive response of pressure sensors with different mass fractions of MWCNTs under 60 kPa. (**c**) Linear fitting of capacitance response for the sensor under 60 kPa. (**d**) Relative capacitive response of the pressure sensor during a 60 kPa hysteresis test. (**e**) Sensor sensitivity and linearity range compared to recently published capacitive pressure sensors. (**f**) Relative capacitive response of the sensor during a step pressure test at 0–60 kPa. (**g**) Response time of the sensor at a pressure of 1.3 kPa. (**h**) The stability test of the sensor for 500 cycles at 60 kPa.

**Figure 8 polymers-16-00962-f008:**
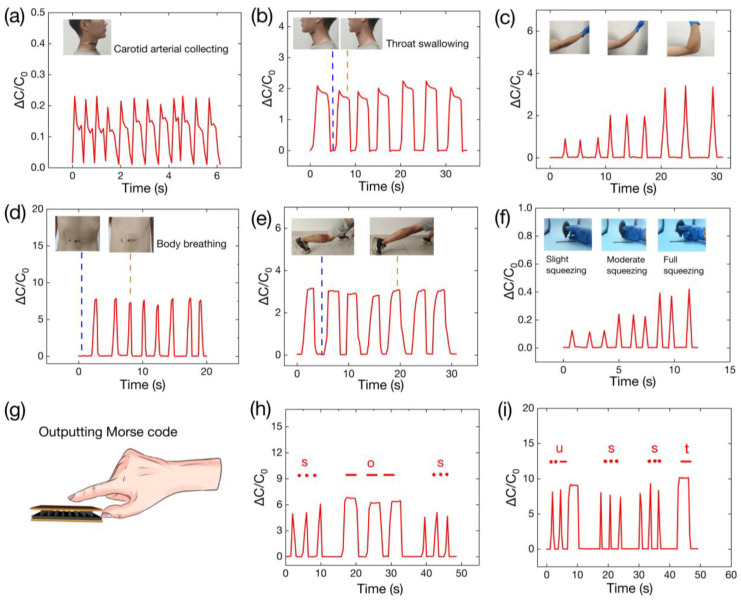
Relevant applications of the sensor. (**a**) The detected signal from carotid artery pulsation when the sensor is bonded to the neck. (**b**) The detection of swallowing signals when the sensor is bonded to the throat surface. (**c**) The sensor attached to arm joints to detect different degrees of flexion. (**d**) The sensor detecting breathing in the human abdomen. (**e**) The detection of muscle activity when the sensor is bonded to the calf. (**f**) Different levels of squeeze applied to an air blow ball result in different degrees of airflow, which can be detected by the sensor. (**g**–**i**) Morse code output from a finger pressing on the sensor.

## Data Availability

Data are contained within the article.

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
