# Peer review of "Linear Capacitive Pressure Sensor with Gradient Architecture through Laser Ablation on MWCNT/Ecoflex Film"

_polymers, 2024, doi:10.3390/polym16070962_

Round 1

Reviewer 1 Report

Comments and Suggestions for Authors

The manuscript presents the fabrication of flexible capacitive pressure sensor based on MWCNT/Ecoflex dielectric layer. Several possible applications have been demonstrated. The discussion and conclusion are sound and supported by the results. In my opinion, it can be accepted for publication after revision as suggested below;

- Authors state that “Specific quantities of MWCNT and dispersant (polyvinylpyrrolidone, PVP) were 89 precisely measured using an electronic scale and combined in a ratio of 10:1”. It’s %wt. or % ?? Why MWCNT and PVP is 10:1? Please give the reason?

- The SEM images (Fig. 3g and 3h) lack clarity. Please enhance them to clearly depict the MWCNT/Ecoflex structure.

- The limit of detection should be investigated.

- Why Fig. 6b does not have the error bar? Please clarify.  

- Is the measurement in Fig. 6d continuous? I believe it's not continuous. Please verify and make the necessary correction.

- Why 2.5 wt% MWCNT give the highest response? Please give the reason.

- In Figure S8, they should be µm not mm. Please correct.

- Based on Figure S8, it’s hard to understand why the dielectric layer thickness of 400 µm gives the linear range. It’s the same material. Please clarify.

- The clear photograph of the finished flexible sensor should be included in the manuscript.

- The English throughout the manuscript should be carefully reviewed.

Comments on the Quality of English Language

 Extensive editing of English language required.

Author Response

We appreciate reviewer for his/her effort to review our manuscript, and his/her positive feedback. The reviewer gives an accurate summary of our work and brings forward constructive questions. We have studied reviewers’ comments carefully and made corresponding revision in the manuscript and supporting information. Attached please find the revised version, which we would like to submit for your kind consideration.

Reviewer 2 Report

Comments and Suggestions for Authors

In the manuscript entitled "Linear capacitive pressure sensor with Gradient Architecture by laser ablation on MWCNT/Ecoflex film”, the authors present a new capacitive pressure sensor that can detect a wide range of body signals such as pulse, swallowing, arm joint flexion signals and calf muscle force.

The main question addressed by the research is to fabricate a MWCNT/Ecoflex pressure sensor with gradient microcones architecture by laser ablation and to test its response to the relative dielectric constant of the MWCNT/Ecoflex film under pressure. In addition, the submitted manuscript provided an analysis of the dielectric constant variations of dielectric layer composite films when a film is under pressure, which has not been considered in other studies.

The topic is interesting and relevant to the field as the development of different types of sensors that convert mechanical deformations into electrical signals is of great interest to the scientific community. Their sensitivity to mechanical deformations and the possibility of obtaining a linear response that can be quantified is of great interest due to their various applications. With the results presented, the materials and methods used, this research further improves the subject area in comparison to other publications.

The manuscript is well organized, the authors have applied the scientific methods and they are adequately described. I do not see any specific improvements that the authors should consider regarding the methodology and testing of the presented pressure sensors.

The results are clearly presented and reproducible due to the details of the methods presented. The results show that in the low pressure stage the gradient microcones increase the sensitivity and linearity of the sensor. In the medium to high pressure stage, the dielectric layer was pressed into a film, and changing the high dielectric constant of the film improves linearity and sensitivity.

In the conclusion part, the authors presented the results of the fabricated sensors and emphasized their advantages in terms of manufacturing efficiency and customizability. The conclusions are consistent with the results presented and the objectives set at the beginning of the work were achieved.

The references cited in the manuscript are recent, mostly from the last 10 years. It should be noted that has to be taken into account that references from 1-25 are only listed and it is not possible to determine why these manuscripts are important to this research without a detailed analysis. It is necessary to briefly announce the content of these references.

The figures, tables, images and data information are presented appropriately and clearly. The data presented in diagrams are appropriately presented and easy to interpret and understand.

The following corrections should be made:

- the beginning of the sentence is missing in lines 32 and 33.

- the note on references should be taken into account.

It is suggested to accept the manuscript with the mentioned corrections.

Author Response

(The authors gave the same response as above.)

Round 2

Reviewer 1 Report

Comments and Suggestions for Authors

The manuscript has been significantly improved. Authors have clarified the issues. Therefore, it can be accepted for publication.

Comments on the Quality of English Language

-